# Association of Hours of Paid Work with Dietary Intake and Quality in Japanese Married Women: A Cross-Sectional Study

**DOI:** 10.3390/nu13093005

**Published:** 2021-08-28

**Authors:** Fumi Oono, Nozomi Matsuura, Aki Saito, Aya Fujiwara, Osamu Takahashi, Satoshi Sasaki, Kaoruko Iida

**Affiliations:** 1Department of Nutrition and Food Science, Graduate School of Humanities and Sciences, Ochanomizu University, 2-1-1 Otsuka, Bunkyo-ku, Tokyo 112-8610, Japan; oonofumi@g.ecc.u-tokyo.ac.jp (F.O.); nozomi.ohara0630@gmail.com (N.M.); aki.saito5@gmail.com (A.S.); 2Department of Social and Preventive Epidemiology, Division of Health Sciences and Nursing, Graduate School of Medicine, The University of Tokyo, 7-3-1 Hongo, Bunkyo-ku, Tokyo 113-0033, Japan; stssasak@m.u-tokyo.ac.jp; 3Department of Nutritional Epidemiology and Shokuiku, National Institutes of Biomedical Innovation, Health and Nutrition, 1-23-1 Toyama, Shinjuku-ku, Tokyo 162-8636, Japan; fujiwaraay@nibiohn.go.jp; 4Department of Social and Preventive Epidemiology, School of Public Health, The University of Tokyo, 7-3-1 Hongo, Bunkyo-ku, Tokyo 113-0033, Japan; 5Department of Epidemiology, Graduate School of Public Health, St. Luke’s International University, 3-6-2 Tsukiji, Chuo-ku, Tokyo 104-0045, Japan; otakahas@luke.ac.jp; 6Department of General Internal Medicine, St. Luke’s International Hospital, 9-1 Akashi-cho, Chuo-ku, Tokyo 104-8560, Japan; 7Institute for Human Life Innovation, Ochanomizu University, 2-1-1 Otsuka, Bunkyo-ku, Tokyo 112-8610, Japan

**Keywords:** employment, dietary intake, housewife, worker, diet quality, Japan

## Abstract

This study investigated the association of hours of paid work with dietary intake and diet quality among Japanese married women. This cross-sectional study was a secondary analysis of a nationwide population survey in 2013. The analytic sample included 644 married women aged 20–59 years. The participants were categorized into five groups according to hours of paid work per week: 0 (housewives), 1–14, 15–34, 35–42, and ≥43 h. Dietary intake was assessed by a self-administered diet history questionnaire. The Nutrient-Rich Food Index 9.3 (NRF9.3) was used to measure the dietary quality. The association of hours of paid work with dietary intake and NRF9.3 score was assessed using a multivariable general linear regression analysis with adjustments for confounders. Hours of paid work were associated with a higher intake of rice and lower intake of vegetables, potatoes, soy products, and seaweeds and nutrients including protein, dietary fiber, and most vitamins and minerals. Hours of paid work were negatively associated with the NRF9.3 score. This study showed that Japanese married women engaging in paid work, especially those who work long hours, have less healthy diets. Efforts to improve the dietary intake of married women with paid work might be needed.

## 1. Introduction

Married women have traditionally been responsible for much of the housework [1]. Throughout the last few decades, in addition to a role in housework, a growing number of married women have also engaged in paid work. For instance, in Japan, the employment rate of married women aged 25–64 years old has considerably increased (from 55% in 2000 to 71% in 2020) [2]. Still, the hours of unpaid work (e.g., housework and child-care) taken up by women are so much longer than those of men, globally [3,4].

Engagement in paid work results in less time for other activities. Previous studies on women and employment reported that their long hours of paid work or full-time work were associated with less time for meal preparation [5,6], which may lead to unhealthy dietary intake [5,7]. In contrast, engagement in paid work provides the benefit of income, which may improve diet quality [8]. Despite the increase in the employment rate of married women, few previous studies have examined the association of engagement in paid work with dietary habits among married women or women considered to have a role in unpaid work [5,9,10,11,12,13]. In particular, the overall dietary intake has not been well examined in relation to engagement in paid work [11,12]. Most previous studies have focused on limited dietary habits, such as breakfast habits [5,9], fast food consumption [5,10,13], and intake of select food groups or beverages [5,13]. Examining overall diet is beneficial in providing useful indications for improving diet and health in married women who engage in paid work or who hold multiple roles.

The association of engagement in paid work with dietary habits can vary among the background of target populations. For women who have a large responsibility for meal preparation, the impact of their hours of paid work on dietary behaviors can be substantial. According to the Global Gender Gap Report in 2018, women across the world spend twice as much time on average performing unpaid work (e.g., housework and child-care) than men; however, women in Japan, Korea, and India spend over fivefold as much time [14]. In Japan, the average daily time spent on meal preparation and dishwashing is 142 min for wives and 6 min for husbands in households with housewives, whereas this time is 110 min for wives and 8 min for husbands in dual-earner households [15]. This finding suggests that women in dual-earner households have less time for meal preparation. No study has yet examined the association between engagement in paid work and dietary intake among married women in the countries mentioned above, except for a study on pregnant women with much lower employment rates [11]. The length of hours of paid work should also be considered, given that time scarcity has been reported to be negatively associated with some aspects of a healthy diet [16,17,18,19,20].

Two main points have been identified as a research gap: (1) diet is rarely comprehensively examined in this subject; (2) no study has been conducted to date in the countries where women perform a large share of the housework. This study aimed to investigate the association of hours of paid work with dietary intake and diet quality among married women in Japan, where women assume a large share of the housework. We hypothesized that the dietary intake of married women with longer hours of paid work is less healthy than that of those with shorter hours of paid work (including housewives).

## 2. Materials and Methods

### 2.1. Study Population

This cross-sectional study is a secondary analysis of the Health Diary Study 2, a nationwide survey in October 2013, which followed the first Health Diary Study conducted in 2003 [21]. The survey aimed to examine the sociodemographics, health events, and health behaviors in a representative Japanese population. The details of the survey have been described elsewhere [22,23,24]. Briefly, potential survey participants were recruited from a Japan Statistics and Research Co., Ltd. research panel. Individuals in the panel were initially contacted primarily through newspaper advertisements. The panel consisted of 79,749 individuals (42,106 women; 37,643 men) aged 0–99 years at the time. After recruitment for the survey via facsimile, 19,633 individuals (10,901 women; 8732 men) agreed to participate. Among these, 5000 individuals (2571 women; 2429 men) were selected based on population-weighted random sampling, controlling for the proportion of sex, age, and size of the residential area to be the same as in the 2010 Japanese National Census [25]. Self-administered questionnaires and prepaid return envelopes were mailed to the 5000 selected individuals. Participants were asked to complete and return the following two questionnaires: (1) a questionnaire about sociodemographic factors, health status, and health-related factors; (2) a brief-type self-administered diet history questionnaire (BDHQ). The BDHQ was only sent to participants aged ≥18 years, in line with the design of the questionnaire. In total, 4548 individuals (2365 women; 2183 men) completed the first questionnaire, and 3700 participants (1968 women; 1732 men) aged ≥18 years completed the BDHQ. Participants who completed the questionnaires received 4000 Japanese yen. The survey protocol was approved by the Ethics Committee of St. Luke’s International Hospital, Tokyo, Japan (No. 13-R91). Written informed consent was obtained from all participants.

For this study, we selected a sample of married women (*n* = 718) from 1163 women aged 20–59 years, considering that the most frequent retirement age in Japan is 60 years [26]. We then excluded those who had received dietary restrictions under the supervision of a dietitian or physician (*n* = 22); who reported being pregnant or lactating (*n* = 11); who were suspected of over- or under-reporting energy intake (whose energy intake was more than 1.5 times the sex- and age-specific estimated energy requirement [EER] for the highest physical activity level or less than half the EER for the lowest physical activity level [*n* = 20]); who responded “neither housewife nor worker” regarding occupational status (for details, see below “2.4 h of paid work”; *n* = 13); those who had missing data in the variables used in the analysis (*n* = 8). The EER and physical activity levels were based on the 2015 Japanese dietary reference intakes (DRIs) [27]. The final analytic sample included 644 married women aged 20–59 years (Figure 1). The sample size was not calculated *a priori* in this study because it was a secondary analysis of the survey.

### 2.2. Dietary Intake

Habitual dietary intake was assessed using the BDHQ. The BDHQ is a 4-page fixed-portion questionnaire that examines the intake frequency of 58 selected food and beverage items during the preceding month [28,29]. The daily food, nutrient, and energy intakes were estimated based on the Standard Tables of Food Composition in Japan [30]. Added sugar intake was similarly calculated based on a recently developed comprehensive composition database by subtracting total sugar contents derived from fruit juices from free sugar contents [31]. According to studies assessing the relative validity of the BDHQ against the 16-day dietary records in Japanese women aged 31–69 years, the median value of Spearman’s correlation coefficients for food intake was 0.44 (range: 0.14–0.82) [28], the corresponding value of Pearson’s correlation coefficients for nutrient intake was 0.54 (range: 0.34–0.87) [29], and the Spearman’s correlation coefficients for added sugar was 0.40 [32]. Food and nutrient intakes were energy-adjusted, using the density method for assessing dietary intake differences because of varied energy requirements and to reduce the influence of misreporting [33]. Nutrient intake from dietary supplements was not considered in the nutrient intake calculation because the information was not obtained in the survey.

### 2.3. Dietary Quality

Dietary quality was assessed using the Nutrient-Rich Food Index 9.3 (NRF9.3). The NRF9.3 represents the nutrient density of the total diet, based on nine qualifying nutrients (protein, dietary fiber, vitamins A, C, and D, calcium, iron, potassium, and magnesium) and three disqualifying nutrients (added sugars, saturated fats, and sodium) [34]. We calculated the NRF9.3 according to a previous study in Japan [35]. Briefly, the NRF9.3 was calculated by subtracting the sum of the share that exceeded percentages of reference daily values (RDVs) for three disqualifying nutrients from the sum of the percentage of RDVs for nine qualifying nutrients. RDVs were derived from the 2015 Japanese DRIs [27] and the conditional recommendation for added sugar advocated by the World Health Organization (<5% of energy) [36] (as shown in Appendix A). The nutrient intake of each participant was adjusted using EER: Adjusted nutrient intake (unit/day) = observed intake (unit/day) × EER (kcal/day)/observed energy intake (kcal/day). The sex- and age-specific EER values for a moderate level of physical activity were derived from the 2015 Japanese DRIs. For qualifying nutrients, the percentage of RDVs was capped at 100% so that high intake of qualifying nutrients would not compensate for the low intake of other qualifying nutrients. A higher NRF9.3 score indicated a better diet quality, with a maximum possible score of 900. The relative validity of the NRF9.3 calculated using the BDHQ has been examined against a 16-day weighed dietary record as a reference in Japanese women aged 31–69 years (Pearson’s correlation coefficient: 0.61) [35].

### 2.4. Hours of Paid Work

Hours of paid work were assessed using two questions: (1) the occupational status, and (2) hours of paid work. Occupational status was categorized as manager or company officer, company employee, public worker, professional, self-employed, part-time worker (these six categories were regarded as a worker), and housewife, based on the responses. Only workers were asked to answer the second question about hours of paid work. Average hours of paid work per week excluding break-time (e.g., lunch break) were selected from the following options: 1–14, 15–34, 35–42, 43–48, 49–59, and ≥60 h. These options were based on a national survey [37] with modifications for the Health Diary Study. Given that only 15 (2.4%) participants answered that they worked for 49–59 h or ≥60 h, we consolidated 43–48, 49–59, and ≥60 h into ≥43 h. For participants who answered housewives, we assigned zero hours per week as their hours of paid work. Thus, we ultimately used five categories: 0 (housewives), 1–14, 15–34, 35–42, and ≥43 h.

### 2.5. Other Variables

In the self-administered questionnaires, participants reported their age, body height, body weight, education level, household income, family members living together, residential area, smoking status, and alcohol habits. Age was categorized as 20–29, 30–39, 40–49, and 50–59 years. Body mass index (BMI) was calculated by dividing the weight (kg) by the square of the height (m). Participants were classified into three categories (<18.5, 18.5–24.9, and ≥25 kg/m^2^) according to the Japan Society for the Study of Obesity [38]. Education was categorized into three levels: low (high school or less), middle (vocational school or junior college), and high (university or higher). Annual household income (sum of the income of all members of the same household, including pensions) before taxation was obtained with 10 response options and categorized into approximate quartiles: ≤4,000,000 yen; 4,000,000 to ≤6,000,000 yen; 6,000,000 to ≤8,000,000 yen; and 8,000,000 yen or more. Due to the lack of information about the number of household members, crude annual household income was used. Family members living together were selected from the following response options: living alone; with spouse only; with parents and a spouse; with a spouse and children; with parents, a spouse, and children; with a spouse, children, and grandchildren; others. Regarding living with children, the response options were grouped into living with or without children (i.e., whether they live with their children or not). Regarding living with parents, the response options were similarly grouped into living with or without parents. Residential areas were classified as a ward (Tokyo metropolitan districts or ordinance-designated cities where the population is more than 500,000); city (the population is more than 50,000); town or village, using their reported postal code. Smoking status was grouped into current smokers or nonsmokers. Alcohol habits were categorized into non-consumers or consumers at least once in the previous month because 48% of the participants (*n* = 309) answered they had not consumed an alcoholic beverage in the previous month.

### 2.6. Statistical Analysis

All the statistical analyses were performed using SPSS (version 24, Chicago, IL, USA). All reported *p* values were 2-tailed, with *p* < 0.05 considered statistically significant. The participant characteristics according to the categories of hours of paid work were compared using Pearson’s chi-squared test or an analysis of variance. When the chi-squared test found a significant association, the standardized residual was calculated. As *post hoc* analysis for a continuous variable, a Tukey–Kramer test was performed.

Multivariable general linear regression models were constructed to examine the association between hours of paid work and dietary intake and NRF9.3 score. Mean values and standard error values of dietary intake and NRF9.3 score were calculated for each category of hours of paid work with adjustments for potential confounders. Trends of the association of hours of paid work with dietary intake and NRF9.3 scores were tested, assigning the median value of hours of paid work in each category as a continuous variable except for ≥43 h. For the category of ≥43 h, the median value of 43–49 h (i.e., 46 h) was assigned. As potential confounders, we included age, education level, household income, living with children, residential area, and smoking status. This study only showed dietary data adjusted by potential confounders because crude and adjusted models showed similar results.

For the sensitivity analyses, since the number of those with ≥43 h was relatively small (*n* = 36), participants were grouped into a total of four groups of hours of paid work: 0, 1–14, 15–34, and ≥35 h. Additionally, analyses including BMI, alcohol habits, energy intake, and living with parents as confounders were performed.

## 3. Results

Table 1 presents characteristics of the 644 married women aged 20–59 years grouped according to the categories of hours of paid work. The percentage of women in 0 (housewives), 1–14, 15–34, 35–42, and ≥43 h of paid work was 37.3%, 12.7%, 30.7%, 13.7%, and 5.6%, respectively. There was a difference in alcohol habits between categories of hours of paid work (*p* = 0.040): the prevalence of alcohol consumers was lower in housewives (44.2%) than in all participants (52.0%) (standardized residual: −3.1). The energy intake was higher in those working ≥43 h than in the other groups (*p* = 0.003, 0.028, 0.015, and 0.046 for 0, 1–14, 15–34, and 35–42 h, respectively). There were no significant differences in the other investigated variables across the categories of hours of paid work.

The mean (standard deviation) scores of the NRF9.3 for all participants was 682 (102) (Appendix A). Among components of the NRF9.3, more than 95% of the participants met the RDVs for protein and vitamin D, whereas the percentages of participants meeting the RDVs for dietary fiber and sodium were less than 10% (8.7% and 0.4%, respectively) (Appendix A).

Table 2 shows the estimated food group intake across categories of hours of paid work. Irrespective of adjustment for potential confounders, rice was positively associated with hours of paid work. The intakes of potatoes, soy products, vegetables, and seaweeds were negatively associated with hours of paid work. Hours of paid work were not significantly associated with intakes of the other food groups.

Nutrient intake across categories of hours of paid work is shown in Table 3. The intakes of protein, dietary fiber, vitamins A, B1, B2, B6, and C, folate, potassium, calcium, magnesium, iron, and copper were negatively associated with hours of paid work, while hours of paid work were not significantly associated with intakes of the other nutrients.

The adjusted mean (standard error) of the NRF9.3 score was 695 (6.5), 688 (11.0), 674 (7.1), 672 (10.7), and 662 (16.7) for 0 (housewives), 1–14, 15–34, 35–42, and ≥43 h, respectively (Table 3). Longer hours of paid work were associated with a lower NRF9.3 score (*p* for trend = 0.006).

The sensitivity analysis confirmed similar results when 35–43 h and ≥43 h were consolidated into ≥35 h (resulting in four categories) or when the model additionally included BMI, alcohol habits, energy intake, and living with parents as potential confounding factors (data not shown).

## 4. Discussion

To our knowledge, this is the first study to investigate the association of hours of paid work with dietary intake among married women in countries where women perform a large share of the housework. The results support our hypothesis that longer hours of paid work would lead to less healthy dietary intake. In the present study, irrespective of the adjustment for potential confounders such as household income, hours of paid work were negatively associated with dietary quality and intake of vegetables, potatoes, soy products, seaweeds, protein, dietary fiber, and many vitamins and minerals, and positively associated with rice intake. The findings of this study should not be interpreted as diminishing the value of women’s participation in the labor force; rather, they indicate a need to improve the dietary quality of married women who engage in paid work.

Few studies have examined the association between engaging paid work and overall dietary intake in women considered to have more roles in unpaid work. Although engaging in paid work in addition to child-care can lead to time scarcity, a study among Australian women with children showed few associations of employment with their nutrient intake [12]. The discrepancies in results with the present study could be partly due to the difference in gender roles in each country, in addition to the difference in the study population. In Japan, women spent more than five times as many hours performing unpaid work as men, whereas women spent less than twice as much time engaged in unpaid work as men in Australia [4]. On the other hand, a study on Japanese pregnant women showed that employment was associated with lower intake of soy products, nuts, dietary fiber, magnesium, iron, vitamin A, and folate [11]. This result is partly consistent with our current results, and the present study additionally observed the negative associations between hours of paid work and the intake of other foods (e.g., vegetables), vitamins, and minerals. The similarity is possibly due to characteristics unique to Japanese women (e.g., diet and gender roles), and the difference could be reflected in the gap in the proportion of women with paid work between the previous study (28.7%) and the present study (62.7%). Moreover, pregnant women are considered to take extra care of their diet. Future studies are needed to confirm the association between hours of paid work of married women and dietary intake.

Monetary affluence is likely one of the factors related to a healthy diet [8], which may explain some of the association between hours of paid work and dietary behaviors. In a study in Pakistan that included women with children, working women had a higher intake of fruits and vegetables compared with housewives [13]. This finding is perhaps because working women had much a higher household income than housewives in that population. In contrast, in a US study, women with full-time work had less fruit and vegetable intake than housewives, despite higher household income among women with full-time work [5]. That study also showed less food preparation time among women with full-time work compared with housewives, which indicates that food preparation time may have had a large impact on their dietary intake rather than income. In the current study, hours of paid work were associated with unhealthy dietary intake, independent of household income. It is possible that time constraints due to paid work may affect dietary intake in Japanese married women. Nevertheless, the association between hours of paid work and dietary habits may vary based on the sociodemographic characteristics of the target population, and consideration of these factors is needed for thorough understanding of the association of hours of paid work with dietary intake.

Lack of time has been reported as one of the major perceived barriers to eating a healthy diet [20,39,40,41]. A lower intake of vegetables and micronutrients, which were associated with hours of paid work in this study, was associated with time scarcity [18,40], and less meal preparation time [7,39]. These factors could explain the association between longer hours of paid work and less healthy dietary intakes in this study. Although these factors have also been reported to be associated with higher fat and fast-food intakes in previous studies [16,39,40,41,42], we did not observe the associations of hours of paid work with fat intake. Further studies are needed to fill the gap in these findings.

We observed negative associations with hours of paid work of intake of dietary fiber and many vitamins and minerals. Lower intakes of these nutrients can be attributed to mainly lower intake of vegetables and, in part, lower intake of potatoes and pulses. On the other hand, rice was positively associated with hours of paid work in this study. In previous Japanese studies, a higher rice intake was associated with lower intakes of other food groups (e.g., vegetables), dietary fiber, and micronutrients [43,44]. Considering that rice is a major staple food in Japan and relatively energy-dense, a higher rice intake co-occurring with a lower micronutrient intake is unsurprising. By examining overall dietary intake, the current study identified foods and nutrients that were more versus less affected by engagement in paid work. In Japan, married women’s engagement in paid work may affect their intake of healthy foods and nutrients, but not their intake of unhealthy ones.

Although several nutrients showed significant associations with hours of paid work whereas others did not, these results should be interpreted with careful consideration of the amount of the nutrient intake in question in this study population. For example, hours of paid work were associated with lower protein intake, whereas most (98.8%) participants met the RDV for protein. An increase in protein intake might not be needed, even in married women who work long hours. Conversely, less than 10% of participants met the RDV for dietary fiber, and the mean dietary fiber intake was much lower than the RDVs regardless of the categories of hours of paid work. An increase in dietary fiber intake is desirable for Japanese married women, especially those with long hours of paid work. Although less than 1% of participants met the RDV for sodium, hours of paid work were not associated with sodium intake. Reducing sodium intake should be encouraged for married women regardless of hours of paid work. Our results showed that women with longer hours of paid work are more likely to have unhealthy diets, but there is still a lot of room to improve dietary intake even in housewives.

The current findings suggest the necessity of establishing an environment where married women who engage in paid work can improve their dietary intake in Japan. Although further research is needed, public policy efforts are also required to establish such an environment. Increasing the availability of healthy and less time-consuming foods may be one of the useful strategies for coping with a lack of time for food preparation [17,45]. Additionally, regardless of sex, it may be essential to control the appropriate hours of paid work. Japanese men spend longer hours in paid work than men in any other OECD country [4], which may make it difficult for men to share the housework with their partners. Although it is not possible to provide effective interventions only based on the current findings, public health strategies may be required to not only encourage individual efforts for improving their dietary intake but also focus on decreasing time burdens among workers.

Several limitations of the study warrant mention. First, we could not obtain the data to describe the share of housework and to explain the observed associations. If meals were mainly prepared by other household members, married women’s hours of paid work would not lead to their unhealthy dietary intake. Although it is difficult to assume that this study population differs considerably in the share of housework taken from the national survey, which showed that married women spend more than ten times as long as men even in dual-earner households in Japan [15], the association might vary depending on the division of housework in each household.

Second, this study was conducted in 2013, and since then the association between hours of paid work and dietary intake for married women may have been changed depending on their social circumstances. Little has changed in the amount of average time that married men and women spent on meal preparation and dishwashing between 2011 [46] and 2016 [15] according to the surveys, whereas the employment rate of married women aged 15−64 years increased from 64% in 2013 to 74% in 2020 [2]. It is difficult to conclude, solely based on the findings of this study, what impact these changes may have had on the association of hours of paid work with dietary intake in married women. Future studies are needed to examine this association at periodic intervals.

Third, our participants might not be a representative sample of married Japanese women, partly because of the design of the survey. Although the proportion of workers aged 20–59 years in this study (62.7%) was similar to the proportion of workers in the population of Japanese married women aged 25–54 years (64%) [2], people with little time might have been less likely to participate in the survey. Moreover, hours of paid work were not significantly associated with education level and household income in this study population, whereas some studies reported the associations between women’s employment and such socioeconomic factors in Japan [47] and in other high-income countries [5,12]. Although we included such socioeconomic factors in the analytical model regardless of their association with hours of paid work, it is difficult to evaluate the effects of the socioeconomic discrepancies on the main findings and generalizability.

Fourth, in this study, hours of paid work were obtained from predetermined categories. The category of 35–42 h was considered to be the typical scheduled hours for full-time work because 96% of companies in Japan had a scheduled work time of 35–40 h [48]. However, it is possible that this categorization was not the best for detecting the association of hours of paid work with dietary intake.

Finally, the magnitude of misreporting of dietary intake can vary by the categories of hours of paid work. The reporting accuracy could be poor among the participants with long hours of paid work, considering a possible association between hours of paid work and frequency of preparing meals on their own. For some nutrients, such as potassium and sodium, assessment using biomarkers could be useful so that results are not due to misreporting.

## 5. Conclusions

This study showed that married women engaging in paid work, especially those with long hours of paid work, have lower dietary quality, lower intake of vegetables, soy products, seaweeds, and potatoes, as well as lower intakes of dietary fiber, and many vitamins and minerals in Japan. These findings suggest that hours of paid work are one of the factors that led to unhealthy dietary intake in married women performing a large share of the housework. Although housewives also have room for improvement in their dietary intake, further research is needed to determine what factors are necessary for married women who engage in paid work to facilitate healthy dietary intake.

## Figures and Tables

**Figure 1 nutrients-13-03005-f001:**
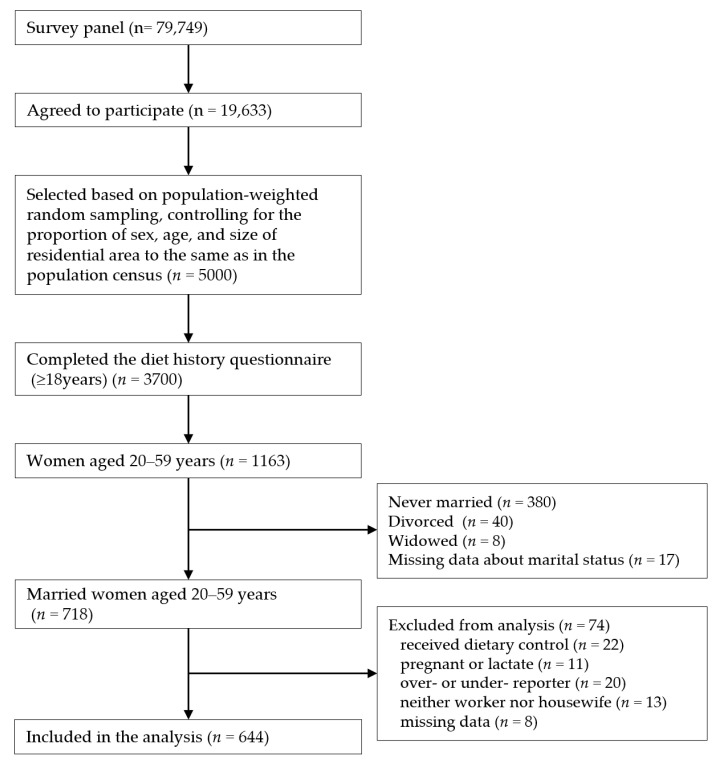
Study participants, comparing the dietary intake of Japanese married women with respect to hours of paid work.

**Table 1 nutrients-13-03005-t001:** Basic characteristics according to the categories of hours of paid work in Japanese married women.

	All		Hours of Paid Work Per Week										*p* ^c^
			0 (Housewives)		1–14		15–34		35–42		≥43		
*n*	644	(100)	240	(37.3)	82	(12.7)	198	(30.7)	88	(13.7)	36	(5.6)	
Age category													0.59
20–29 years	10	(1.6)	4	(1.7)	0	(0.0)	2	(1.0)	3	(3.4)	1	(2.8)	
30–39 years	177	(27.5)	73	(30.4)	20	(24.4)	53	(26.8)	25	(28.4)	6	(16.7)	
40–49 years	221	(34.3)	72	(30.0)	32	(39.0)	69	(34.8)	33	(37.5)	15	(41.7)	
50–59 years	236	(36.6)	91	(37.9)	30	(36.6)	74	(37.4)	27	(30.7)	14	(38.9)	
BMI category													0.29
Underweight: <18.5 kg/m^2^	92	(14.3)	40	(16.7)	13	(15.9)	25	(12.6)	10	(11.4)	4	(11.1)	
Normal weight: 18.5–24.9 kg/m^2^	469	(72.8)	173	(72.1)	58	(70.7)	152	(76.8)	59	(67.0)	27	(75.0)	
Overweight: ≥25 kg/m^2^	83	(12.9)	27	(11.3)	11	(13.4)	21	(10.6)	19	(21.6)	5	(13.9)	
Education level													0.59
High school or less	228	(35.4)	82	(34.2)	26	(31.7)	74	(37.4)	32	(36.4)	14	(38.9)	
Vocational school or junior college	284	(44.1)	103	(42.9)	45	(54.9)	84	(42.4)	36	(40.9)	16	(44.4)	
University or higher	132	(20.5)	55	(22.9)	11	(13.4)	40	(20.2)	20	(22.7)	6	(16.7)	
Household income (Japanese yen)													0.44
<4,000,000	117	(18.2)	53	(22.1)	15	(18.3)	29	(14.6)	13	(14.8)	7	(19.4)	
≥4,000,000 & <6,000,000	190	(29.5)	72	(30.0)	23	(28.0)	66	(33.3)	20	(22.7)	9	(25.0)	
≥6,000,000 & <8,000,000	161	(25.0)	52	(21.7)	24	(29.3)	53	(26.8)	23	(26.1)	9	(25.0)	
≥8,000,000	176	(27.3)	63	(26.3)	20	(24.4)	50	(25.3)	32	(36.4)	11	(30.6)	
Living with children	424	(65.8)	161	(67.1)	58	(70.7)	133	(67.2)	55	(62.5)	17	(47.2)	0.13
Living with parents	71	(11.0)	22	(9.2)	8	(9.8)	18	(9.1)	17	(19.3)	6	(16.7)	0.059
Size of residential area													0.16
Ward	192	(29.8)	82	(34.2)	22	(26.8)	50	(25.3)	24	(27.3)	14	(38.9)	
City	413	(64.1)	149	(62.1)	52	(63.4)	135	(68.2)	59	(67.0)	18	(50.0)	
Town & village	39	(6.1)	9	(3.8)	8	(9.8)	13	(6.6)	5	(5.7)	4	(11.1)	
Current smokers	36	(5.6)	8	(3.3)	3	(3.7)	14	(7.1)	7	(8.0)	4	(11.1)	0.50
Alcohol consumers ^a^	335	(52.0)	106 ^e^	(44.2)	46	(56.1)	110	(55.6)	53	(60.2)	20	(55.6)	0.040
Energy intake (kcal/day) ^b^	1658 ± 430		1623 ± 410		1649 ± 466		1657 ± 396		1666 ± 454		1900 ^d^ ± 541		0.011

All values are the number (percentage) of participants except for energy intake. ^a^ Participants who drank alcohol at least once in the previous month were regarded as consumers. ^b^ Values are mean ± standard deviation. ^c^ Pearson’s X^2^ test was performed for categorical variables, and analysis of variance was performed for continuous variables (i.e., energy intake). ^e^ As *post hoc* analysis, the standardized residual was calculated, showing the prevalence of alcohol consumers was lower in housewives than in all participants. ^d^ As *post hoc* analysis, a Tukey–Kramer test was performed, showing energy intake was higher in those working ≥43 h than in the other groups (*p* = 0.003, 0.028, 0.015, and 0.046 for 0, 1–14, 15–34, and 35–42 h, respectively).

**Table 2 nutrients-13-03005-t002:** Food group intake according to the categories of hours of paid work in 644 Japanese married women.

	Hours of Paid Work Per Week										*p* ^a^
	0 (Housewives)		1–14		15–34		35–42		≥43		
	(*n* = 240)		(*n* = 83)		(*n* = 198)		(*n* = 88)		(*n* = 36)		
(g/1000 kcal)	Mean	SE	Mean	SE	Mean	SE	Mean	SE	Mean	SE	
Rice	134	4.1	131	7.1	146	4.5	156	6.8	132	10.7	0.014
Bread	28.3	1.1	30.6	1.9	27.3	1.2	25.4	1.8	27.0	2.9	0.14
Noodles	37.0	1.5	33.9	2.6	35.2	1.7	39.9	2.5	35.9	3.9	0.68
Potatoes	27.3	1.2	26.0	2.0	25.3	1.3	24.0	2.0	18.7	3.0	0.015
Soy products	42.9	1.6	40.0	2.7	39.9	1.7	35.7	2.6	33.7	4.1	0.007
Vegetables	135	4.0	125	6.9	120	4.4	115	6.7	114	10.4	0.002
Mushrooms	7.7	0.4	6.7	0.6	6.9	0.4	7.1	0.6	6.5	0.9	0.14
Seaweeds	7.0	0.4	7.8	0.7	6.7	0.4	5.7	0.6	5.7	1.0	0.031
Fruit	42.6	2.2	35.8	3.7	40.1	2.4	39.6	3.6	43.2	5.6	0.74
Fish	41.2	1.3	38.9	2.3	36.5	1.4	39.6	2.2	37.8	3.4	0.11
Meat	39.4	1.0	41.0	1.8	39.3	1.1	41.4	1.7	43.3	2.7	0.30
Eggs	21.0	0.9	22.1	1.5	22.6	0.9	20.3	1.4	20.6	2.2	0.98
Dairy products	78.4	3.8	76.0	6.4	78.1	4.1	76.8	6.2	60.6	9.7	0.38
Confectioneries	42.9	1.6	42.5	2.8	43.7	1.8	38.1	2.7	47.1	4.3	0.73
Soft drinks	27.5	3.2	35.4	5.4	32.5	3.5	30.1	5.2	27.2	8.1	0.71

SE, standard error. All values were adjusted for age, education level, household income, living with children, residential area, and smoking status. ^a^ A linear trend test was used with the median value in each category (0, 8, 25, 39, and 46 h for 0 [housewives], 1–14, 15–34, 35–42, and ≥43 h, respectively) as a continuous variable in the multivariable general linear regression.

**Table 3 nutrients-13-03005-t003:** Nutrient intake and NRF9.3 score according to the categories of hours of paid work in 644 Japanese married women.

	Hours of Paid Work Per Week										*p* ^a^
	0 (Housewives)		1–14		15–34		35–42		≥43		
	(*n* = 240)		(*n* = 83)		(*n* = 198)		(*n* = 88)		(*n* = 36)		
	Mean	SE	Mean	SE	Mean	SE	Mean	SE	Mean	SE	
Nutrients											
Protein (% energy)	15.5	0.2	15.2	0.3	15.0	0.2	15.1	0.3	14.7	0.4	0.027
Total fat (% energy)	28.5	0.3	28.6	0.5	28.1	0.3	27.3	0.5	28.2	0.8	0.10
SFA (% energy)	7.9	0.1	8.1	0.2	7.9	0.1	7.5	0.2	7.8	0.3	0.27
Carbohydrate (% energy)	52.8	0.4	51.5	0.8	53.9	0.5	54.0	0.7	51.9	1.1	0.14
Added sugar (% energy)	5.8	0.2	6.3	0.3	6.1	0.2	5.8	0.3	5.9	0.5	0.87
Alcohol (g/1000 kcal)	3.2	0.4	5.2	0.8	2.6	0.5	3.3	0.7	5.7	1.1	0.89
Total dietary fiber (g/1000 kcal)	6.9	0.1	6.5	0.2	6.5	0.1	6.3	0.2	6.0	0.3	<0.001
Vitamin A (µg RAE/1000 kcal)	400	11.3	414	19.2	377	12.3	378	18.6	334	29.0	0.021
Vitamin B1 (mg/1000 kcal)	0.44	0.01	0.43	0.01	0.43	0.01	0.42	0.01	0.42	0.01	0.002
Vitamin B2 (mg/1000 kcal)	0.75	0.01	0.74	0.02	0.73	0.01	0.72	0.02	0.69	0.03	0.026
Niacin (mg NE/1000 kcal) ^b^	16.1	0.2	16.0	0.3	15.5	0.2	15.8	0.3	15.7	0.5	0.14
Vitamin B6 (mg/1000 kcal)	0.70	0.01	0.69	0.02	0.66	0.01	0.67	0.02	0.67	0.03	0.006
Vitamin B12 (µg/1000 kcal)	5.2	0.1	5.2	0.2	4.8	0.2	5.0	0.2	4.9	0.4	0.12
Folate (µg/1000 kcal)	189	3.6	180	6.1	176	3.9	170	5.9	164	9.3	<0.001
Vitamin C (mg/1000 kcal)	64.8	1.6	57.6	2.7	58.2	1.7	56.9	2.6	56.8	4.1	0.003
Vitamin D (mg/1000 kcal)	7.3	0.2	7.0	0.4	6.6	0.3	6.9	0.4	6.7	0.6	0.09
Sodium (mg/1000 kcal)	2344	27	2232	46	2288	29	2281	44	2244	69	0.15
Potassium (mg/1000 kcal)	1472	22	1407	37	1387	24	1356	36	1313	56	<0.001
Calcium (mg/1000 kcal)	318	5.8	307	10.0	302	6.4	293	9.6	273	15.1	0.001
Magnesium (mg/1000 kcal)	143	1.8	138	3.0	136	1.9	134	2.9	131	4.6	<0.001
Iron (mg/1000 kcal)	4.4	0.1	4.3	0.1	4.2	0.1	4.1	0.1	4.0	0.2	<0.001
Zinc (mg/1000 kcal)	4.5	0.03	4.4	0.1	4.4	0.04	4.4	0.1	4.3	0.1	0.11
Copper (mg/1000 kcal)	0.63	0.01	0.60	0.01	0.62	0.01	0.60	0.01	0.58	0.02	0.009
Diet quality score											
NRF9.3	695	6.5	688	11.0	674	7.1	672	10.7	662	16.7	0.006

NRF9.3, Nutrient-Rich Food Index 9.3; SE, standard error; SFA, saturated fatty acid; RAE, retinol activity equivalent; NE, niacin equivalent. All values were adjusted for age, education level, household income, living with children, residential area, and smoking status. ^a^ A linear trend test was used with the median value in each category (0, 8, 25, 39, and 46 h for 0 [housewives], 1–14, 15–34, 35–42, and ≥43 h, respectively) as a continuous variable in the multivariable general linear regression. ^b^ Niacin equivalents were calculated as niacin (mg) + protein (mg)/6000 according to the Dietary Reference Intakes for the Japanese, 2015 [25].

## Data Availability

The data presented in this study are not publicly available. They will be made available upon reasonable request to the corresponding author.

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
