# Peer review of "Association of Hours of Paid Work with Dietary Intake and Quality in Japanese Married Women: A Cross-Sectional Study"

_nutrients, 2021, doi:10.3390/nu13093005_

Round 1

Reviewer 1 Report

The paper  refers to the association between working time and dietary intake /  food quality  among Japanese married women.

The authors may consider following comments:

The work is technically, statistically and linguistically properly written. However, some points evoke my concern.

  1. What was the assumption of the presented division into groups depending on the number of working hours? The division is not equal, 1-14 includes 14 hours, 15-34 includes 19 hours, 25-42 includes only 7. is it a predetermined division? full time, part time etc? if the  working time is continuous variable I would suggest to divide the groups by percentile.
  2. Line 53: “Although the engagement of paid work by married women can affect their dietary 53 behaviors, whether their diet is less healthy than housewives is not understood well”- I do not exactly get the point of that sentence.
  3. Paragraph 2.4. The authors  presented the list of  Is there any difference in food intake and its quality among them?
  4. Table 1. For dichotomized  variables I suggest to present only one line- i.e. smokers,  its obvious that the rest of group represent non-smokers.  Table will be shorter and clearer.
  5. Table 1 presents shows qualitative data except the last line, which is quantitative. I suggest to remove parentheses and use standardized symbol ± .
  6. Table S1 is not attached to the document. If is mentioned in text decide whether remove the references to this table and present the calculations or attach the table.

Summary:  The most important conclusion of the paper is the fact that longer working time is associated with poorer quality of the diet and more carbohydrate consumption. The conclusions seem to be quite simple - the greater amount of work most often results from financial difficulties, which will directly translate into the quality and quantity of groceries as well as the time spent on preparing meals. In this context, the work does not bring much novelty to modern science.

Author Response

August 24, 2021

Reply to the Reviewers’ comments for the manuscript: nutrients-1274960

(Title: Association of hours of paid work with dietary intake and quality in Japanese married women: A cross-sectional study)

We thank the reviewers for their insightful comments on our paper. We have revised the manuscript by addressing each comment point-by-point as shown below, with all amendments in the text highlighted using a red font. We hope these changes will resolve any confusion and remedy the shortcomings of the paper.

Author's Reply to the Review Report (Reviewer 1)

Comment:

The work is technically, statistically and linguistically properly written. However, some points evoke my concern.

Reply:

We appreciate your profound understanding of our work. All of your comments have been very helpful in improving our manuscript and we have revised the manuscript accordingly.

Comment 1:

1.What was the assumption of the presented division into groups depending on the number of working hours? The division is not equal, 1-14 includes 14 hours, 15-34 includes 19 hours, 25-42 includes only 7. is it a predetermined division? full time, part time etc? if the working time is continuous variable I would suggest to divide the groups by percentile.

Reply:

Thank you for your comments; however, the categories of hours of paid work were predetermined, and we could not examine the association hours of paid work and dietary intake based on the categorization using the percentile approach.

In this study, the participants provided information about their average hours of paid work from the following predetermined options: 1–14, 15–34, 35–42, 43–48, 49–59, and ≥60 hours per week. These options were based on the questionnaire of the national survey on time use and leisure activities [37] and modified for Health Diary Study. The hours of paid work did not necessarily correspond to how the employment of the individual was classified: the occupational status, such as part-time worker, company employee, or public worker, was self-defined.

The current study was a secondary analysis of a nationwide survey; therefore, it was difficult to change the predetermined categories of work hours. However, we considered the category to fairly describe the hours of typical full-time work and shorter/longer hours than typical full-time work in Japan. For example, according to a national survey in 2019, the percentages of companies with a scheduled work time of less than 35 hours, 35-39 hours, 40 hours, and 41 hours or more were 0.7%, 32.2%, 63.9%, and 3.3%, respectively [48]. The category of 35-42 hours, which the current study used, is considered to be the typical scheduled hours for full-time work. Still, it is possible that this categorization was not best for the assessment in detecting the association between hours of paid work and dietary intake. To address this limitation, we have added information about the categorization of working hours in the Methods and Discussion sections.

L165-177 (2.4 Hours of paid work)

Hours of paid work were assessed using two questions: (1) the occupational status, and (2) hours of paid work. Occupational status was categorized as manager or company officer, company employee, public worker, professional, self-employed, part-time worker (these six categories were regarded as a worker), and housewife, based on the responses. Only workers were asked to answer the second question about hours of paid work. Average hours of paid work per week excluding break time (e.g., lunch break) were selected from the following options: 1–14, 15–34, 35–42, 43–48, 49–59, and ≥60 hours. These options were based on a national survey [37] with modification for the Health Diary Study. Given only 15 (2.4%) participants answered that they worked for 49–59 hours or ≥60 hours, we consolidated 43–48, 49–59, and ≥60 hours into ≥43 hours. For participants who answered housewives, we assigned zero hours per week as their hours of paid work. Thus, we ultimately used five categories: 0 (housewives), 1–14, 15–34, 35–42, and ≥43 hours.

L406-410(4. Discussion)

Fourth, in this study, hours of paid work were obtained from predetermined categories. The category of 35-42 hours was considered to be the typical scheduled hours for full-time work because 96% of companies in Japan had a scheduled work time of 35-40 hours [48]. Still, it is possible that this categorization was not best for the assessment in detecting the association of hours of paid work with dietary intake.

[37] Statistics Bureau, Ministry of Internal Affairs and Communications. Survey on Time Use and Leisure Activities, Questionnaire B. Available online:

http://www.stat.go.jp/data/shakai/2001/pdf/chob.pdf (accessed on Aug 15, 2021) (In Japanese).

[48] Statistics Bureau, Ministry of Internal Affairs and Communications. General Survey on Working Conditions2020, Table 3, Percentage of companies by company size, industry, and main class of weekly scheduled work hours, and average weekly scheduled work hours. Available online: https://www.e-stat.go.jp/dbview?sid=0003243485 (accessed on Aug 15, 2021) (In Japanese).

Comment 2:

2. Line 53: “Although the engagement of paid work by married women can affect their dietary 53 behaviors, whether their diet is less healthy than housewives is not understood well”- I do not exactly get the point of that sentence.

Reply:

We apologize for the insufficient information. We have revised the Introduction to make the background clear and easy to understand.

L43-61 (1. Introduction)

Married women have traditionally been responsible for much of the housework [1]. Throughout the last few decades, in addition to a role in housework, a growing number of married women have also engaged in paid work. For instance, in Japan, the employment rate of married women aged 25–64 years old has considerably increased (from 55% in 2000 to 71% in 2020) [2]. Still, the hours of unpaid work (e.g., housework and child care) taken up by women are so much longer than those by men globally [3,4].

Engagement in paid work results in less time for other activities. Previous studies on women and employment reported that their long hours of paid work or full-time work were associated with less time for meal preparation [5,6], which may lead to unhealthy dietary intake [5,7]. In contrast, engagement in paid work provides the benefit of income, which may improve diet quality [8]. Despite the increase in the employment rate of married women, few previous studies have examined the association of engagement in paid work with dietary habits among married women or women considered to have a role in unpaid work [5,9-13]. In particular, the overall dietary intake has not been well examined in relation to engagement in paid work [11,12]. Most previous studies have focused on limited dietary habits, such as breakfast habits [5,9], fast food consumption [5,10,13], and intake of select food groups or beverages [5,13]. Examining overall diet is beneficial to pro-vide useful indications for improving diet and health in married women who engage in paid work or who hold multiple roles.

Comment 3:

3. Paragraph 2.4. The authors presented the list of Is there any difference in food intake and its quality among them?

Reply:

We assumed that the intent of the comment was “Paragraph 2.4. The authors presented the list of occupational status. Is there any difference in food intake and its quality among them?” and we have prepared our response given below accordingly. If this understanding is not what you meant, we would appreciate it if you could clarify the question.

Table A shows the comparison of food intake and diet quality according to occupational status. Due to the small number of participants in the categories for manager or company officer, company employee, public worker, professional, and self-employed (7, 56, 20, 6, 34, respectively), we examined the associations using the following three categories; 1) housewives, 2) part-time workers, and 3) other workers (including manager or company officer, company employee, public worker, professional, and self-employed).

Table A. Food group intake and NRF9.3 score according to the categories of occupational status in 644 Japanese married women.

(g/1000kcal)

Occupational status

pa

Housewife

Part time workers

Other workers

(n = 240)

(n = 83)

(n = 198)

Mean

SE

Mean

SE

Mean

SE

Rice

134

4.2

144

3.9

142

5.9

0.14

Bread

28.3

1.1

28.5

1.0

25.4

1.6

0.24

Noodles

37.0

1.5

35.3

1.4

37.7

2.1

0.57

Potatoes

27.3

1.2

24.7

1.1

24.3

1.7

0.19

Soy products

42.9 A

1.6

39.5 A,B

1.5

36.3 B

2.2

0.044

Vegetables

135 A

4.0

120B

3.7

119 A,B

5.7

0.011

Mushroom

7.7

0.4

6.9

0.3

6.9

0.5

0.16

Seaweed

7.0

0.4

6.8

0.4

6.2

0.5

0.43

Fruit

42.6

2.2

38.5

2.0

41.3

3.1

0.38

Fish

41.2

1.3

37.3

1.2

38.8

1.9

0.09

Meat

39.4

1.0

40.6

1.0

40.3

1.5

0.71

Egg

21.0

0.9

21.7

0.8

22.0

1.2

0.76

Dairy products

78.4

3.8

75.4

3.5

76.8

5.3

0.85

Confectioneries

42.9

1.7

42.1

1.5

43.6

2.3

0.85

Soft drinks

27.5

3.1

32.7

2.9

30.7

4.5

0.49

NRF9.3

695

6.5

676

6.0

674

9.1

0.059

NRF9.3b

684

10.0

680

6.7

686

12.1

0.84

All values were adjusted for age, education level, household income, living with children, residential area, and smoking status.
a P values were obtained by analysis of covariances (ANCOVA).

b The values were further adjusted by hours of paid work.
A,B,C Different superscripts within the same row denote statistically significant difference between groups after adjusted using Bonferroni.

NRF9.3, Nutrient-Rich Food Index 9.3.

In summary, occupational status showed weak evidence for the association with diet quality (NRF9.3) (p = 0.059). When the model further included hours of paid work, however, the association was attenuated (please see the row for NRF9.3b) (p = 0.84).

In the current study, our main purpose was to examine the association between hours of paid work and dietary intake based on the hypothesis that the lack of time may be an important factor for dietary intake. Although the questionnaire included a question about occupational status, it is noted that hours of paid work and occupational status may have different meanings. For reference, Table B presents the association of hours of paid work and occupational status in the current study sample. Occupational status did not necessarily describe the duration of time that the participants actually worked: not all part-time workers worked less than 35 hours, and not all other workers (manager or company officer, company employee, public worker, professional, and self-employed) worked more than 35 hours. Regardless of the type of employment, some workers may have extended/shortened the hours of their work for some reason. Therefore, considering the primary purpose of the study, we concluded that it was reasonable to use the hours of paid work for this study, and thus only present the results obtained using working hours.

Table B. Association of hours of paid work and occupational status.

Occupational status

Housewives

Part-time workers

Other workers

Sum

Hours of paid work

Housewives

240 (100%)

0

0

240

1-14

0

69 (84.1%)

13 (15.9%)

82

15-34

0

168 (84.8%)

30 (15.9%)

198

35-42

0

35 (39.8%)

53 (60.2%)

88

>=43

0

9 (25.0%)

27 (75.0%)

36

Comment 4:

4. Table 1. For dichotomized variables I suggest to present only one line- i.e. smokers, its obvious that the rest of group represent non-smokers. Table will be shorter and clearer.

Reply:

Thank you for your suggestion. As you suggested, we have revised Table 1.

(Please see the reply to Comment 5.)

Comment 5:

5. Table 1 presents shows qualitative data except the last line, which is quantitative. I suggest to remove parentheses and use standardized symbol ±.

Reply:

In accordance with your comment, we have revised Table 1. While revising, we noticed that the numbers for women living with parents and without parents were incorrect. We have corrected this error as well as edited the format as suggested in Comment 4.

Table 1. Basic characteristics according to the categories of hours of paid work in Japanese married women.

All

Hours of paid work per week

0
(Housewives)

1–14

15–34

35–42

≥43

pc

n

644

(100)

240

(37.3)

82

(12.7)

198

(30.7)

88

(13.7)

36

(5.6)

Age category

0.59

  20–29 years

10

(1.6)

4

(1.7)

0

(0.0)

2

(1.0)

3

(3.4)

1

(2.8)

  30–39 years

177

(27.5)

73

(30.4)

20

(24.4)

53

(26.8)

25

(28.4)

6

(16.7)

  40–49 years

221

(34.3)

72

(30.0)

32

(39.0)

69

(34.8)

33

(37.5)

15

(41.7)

  50–59 years

236

(36.6)

91

(37.9)

30

(36.6)

74

(37.4)

27

(30.7)

14

(38.9)

BMI category

0.29

Underweight: <18.5 kg/m2

92

(14.3)

40

(16.7)

13

(15.9)

25

(12.6)

10

(11.4)

4

(11.1)

  Normal weight: 18.5–24.9 kg/m2

469

(72.8)

173

(72.1)

58

(70.7)

152

(76.8)

59

(67.0)

27

(75.0)

  Overweight: ≥25 kg/m2

83

(12.9)

27

(11.3)

11

(13.4)

21

(10.6)

19

(21.6)

5

(13.9)

Education level

0.59

  High school or less

228

(35.4)

82

(34.2)

26

(31.7)

74

(37.4)

32

(36.4)

14

(38.9)

  Vocational school or

junior college

284

(44.1)

103

(42.9)

45

(54.9)

84

(42.4)

36

(40.9)

16

(44.4)

  University or higher

132

(20.5)

55

(22.9)

11

(13.4)

40

(20.2)

20

(22.7)

6

(16.7)

Household income (Japanese yen)

0.44

  <4,000,000

117

(18.2)

53

(22.1)

15

(18.3)

29

(14.6)

13

(14.8)

7

(19.4)

≥4,000,000 & <6,000,000

190

(29.5)

72

(30.0)

23

(28.0)

66

(33.3)

20

(22.7)

9

(25.0)

≥6,000,000 & <8,000,000

161

(25.0)

52

(21.7)

24

(29.3)

53

(26.8)

23

(26.1)

9

(25.0)

≥8,000,000

176

(27.3)

63

(26.3)

20

(24.4)

50

(25.3)

32

(36.4)

11

(30.6)

Living with children

424

(65.8)

161

(67.1)

58

(70.7)

133

(67.2)

55

(62.5)

17

(47.2)

0.13

Living with parents

71

(11.0)

22

(9.2)

8

(9.8)

18

(9.1)

17

(19.3)

6

(16.7)

0.059

Size of residential area

0.16

  Ward

192

(29.8)

82

(34.2)

22

(26.8)

50

(25.3)

24

(27.3)

14

(38.9)

  City

413

(64.1)

149

(62.1)

52

(63.4)

135

(68.2)

59

(67.0)

18

(50.0)

  Town & village

39

(6.1)

9

(3.8)

8

(9.8)

13

(6.6)

5

(5.7)

4

(11.1)

Current smokers

36

(5.6)

8

(3.3)

3

(3.7)

14

(7.1)

7

(8.0)

4

(11.1)

0.50

Alcohol consumersa

335

(52.0)

106

(44.2)

46

(56.1)

110

(55.6)

53

(60.2)

20

(55.6)

0.040

Energy intake (kcal/day)b

1658 ± 430

1623 ± 410

1649 ± 466

1657 ± 396

1666 ± 454

1900d ± 541

0.011

All values are number (percentage) of participants except for energy intake.

a Participants who drank alcohol at least once in the previous month were regarded as consumers.

b Values are mean ± standard deviation.

c Pearson's X2 test was performed for categorical variables, and analysis of variance was performed for continuous variables (i.e., energy intake).

e As post hoc analysis, the standardized residual was calculated, showing the prevalence of alcohol consumers was lower in housewives than in all participants.

d As post hoc analysis, a Tukey–Kramer test was performed, showing energy intake was higher in those working ≥43 hours than in the other groups (p = 0.003, 0.028, 0.015, and 0.046 for 0, 1–14, 15–34, and 35–42 hours, respectively).

Comment 6:

6. Table S1 is not attached to the document. If is mentioned in text decide whether remove the references to this table and present the calculations or attach the table.

Reply:

We apologize for the error made by us while uploading the files. We have attached the supplemental material (Table S1).

Table S1. Reference daily values used for the calculation of the NRF9.3 for Japanese women aged 20-59 yearsa and component and total scores of NRF9.3 in 644 Japanese married women.

Reference daily values

NRF9.3 score

n (%) of participants meeting RDVs

20–29 years

30–49 years

50–59 years

Mean

SD

Total NRF9.3 score

682

102

Energy (kcal)b

1950

2000

1900

Qualifying nutrients

Protein (g)c

50

50

50

99.9

0.7

636 (98.8%)

Dietary fiber (g)d

18

18

18

72

17

56 (8.7%)

Vitamin A (μgRAE)c

650

700

700

86

18

 —

Vitamin C (mg)c

100

100

100

91

16

396 (61.5%)

Vitamin D (mg)e

5.5

5.5

5.5

98

9

613 (95.2%)

Calcium (mg)c

650

650

650

85

17

261 (40.5%)

Iron (mg)c

10.5

10.5

6.5

85

16

254 (39.4%)

Potassium (mg)d

2600

2600

2600

94

10

396 (61.5%)

Magnesium (mg)c

270

290

290

90

11

235 (36.5%)

Disqualifying nutrients

Added sugars (g)f

24.4

25.0

23.8

34

51

262 (40.7%)

Saturated fats (g)dg

15.2

15.6

14.8

18

21

189 (29.3%)

Sodium (g NaCl equivalent)dh

7

7

7

66

30

4 (0.6%)

NRF9.3, Nutrient-Rich Food Index 9.3; SD, standard deviation; RDVs, reference daily values; RAE, retinol activity equivalent.
a Values were derived from the Dietary Reference Intakes for Japanese, 2015 [1], except for added sugar, determined based on the World Health Organization's conditional recommendation [2].
b Estimated Energy Requirement for moderate level of physical activity.
c Recommended Dietary Allowance.
d Tentative dietary goal for preventing lifestyle-related disease.
e Adequate intake.
f The values were calculated by 5% of EER (World Health Organization's conditional recommendation).
g The values were calculated by 7% of EER (tentative dietary goal for preventing lifestyle-related disease).
h 7 g NaCl equivalent (g) = 2756 mg sodium.

References
[1] Ministry of Health, Labour and Welfare, Japan. Dietary Reference Intakes for Japanese, 2015. Available online: http://www.mhlw.go.jp/stf/seisakunitsuite/bunya/0000208970.html (accessed on Aug 15, 2021).
[2] World Health Organization. Guideline: Sugars intake for adults and children. Geneva: World Health Organization; 2015. Available online: http://apps.who.int/iris/bitstream/10665/149782/1/9789241549028_eng.pdf?ua=1 (accessed on Aug 15, 2021).

Comment 7:

Summary: The most important conclusion of the paper is the fact that longer working time is associated with poorer quality of the diet and more carbohydrate consumption. The conclusions seem to be quite simple - the greater amount of work most often results from financial difficulties, which will directly translate into the quality and quantity of groceries as well as the time spent on preparing meals. In this context, the work does not bring much novelty to modern science.

Reply:

Thank you for noting this important point. We agree that financial difficulties are important in discussing diet quality. Based on our study outcome, we concluded that the association of hours of paid work and dietary quality was independent of financial difficulties for the following two reasons:

1) Household income was not associated with paid work hours (Table 1)

2) We included household income in the multivariable linear regression models. Irrespective of the adjustment of household income, hours of paid work were associated with lower dietary quality (before adjustment [crude model]: p = 0.011, after adjustment [Table 3]: p = 0.006).

To make the strength of our work clearer, we have added sentences in the Discussion section.

L317-332 (4. Discussion)

Monetary affluence is likely one of the factors related to a healthy diet [8], which may explain some of the association between hours of paid work and dietary behaviors. In a study in Pakistan of women with children, working women had a higher intake of fruits and vegetables compared with housewives [13]. This finding is perhaps because working women had a much higher household income than housewives in that population. In contrast, in a US study, women with full-time work had less fruit and vegetable intake than housewives, despite higher household income among women with full-time work than housewives [5]. That study also showed less food preparation time among women with full-time work compared with housewives, which indicates that food preparation time may have had a large impact on their dietary intake rather than income in that population. In the current study, hours of paid work were associated with unhealthy dietary intake, independent of household income. It is possible that time constraints due to paid work may affect dietary intake in Japanese married women. Nevertheless, the association between hours of paid work and dietary habits may vary based on the sociodemographic characteristics of the target population, and considering these factors is needed for thorough understanding of the association of hours of paid work with dietary intake.

[8] Darmon, N.; Drewnowski, A. Does social class predict diet quality? Am. J. Clin. Nutr. 2008, 87, 1107-1117.

[13] Raza, L.; Ali MT.; Hasnain, A. Comparison Of Dietary Practices And Body Mass Index Among Educated Housewives And Working Women In Karachi. J. Ayub. Med. Coll. Abbottabad 2017, 29, 293-297.

[5] Bauer, K.W.; Hearst, M.O.; Escoto, K.; Berge, J.M.; Neumark-Sztainer, D. Parental employment and work-family stress: Associations with family food environments. Soc. Sci. Med. 2012, 75, 496–504.

While revising, we noticed that some values in Table 3 were incorrect. We have corrected this error (red font).

Table 3. Nutrient intake and NRF9.3 score according to the categories of hours of paid work in 644 Japanese married women.

Hours of paid work per week

0
(Housewive)s

1–14

15–34

35–42

≥43

pa

(n = 240)

(n = 83)

(n = 198)

(n = 88)

(n = 36)

Mean

SE

Mean

SE

Mean

SE

Mean

SE

Mean

SE

Nutrients

Protein (% energy)

15.5

0.2

15.2

0.3

15.0

0.2

15.1

0.3

14.7

0.4

0.027

Total fat (% energy)

28.5

0.3

28.6

0.5

28.1

0.3

27.3

0.5

28.2

0.8

0.10

SFA (% energy)

7.9

0.1

8.1

0.2

7.9

0.1

7.5

0.2

7.8

0.3

0.27

Carbohydrate (% energy)

52.8

0.4

51.5

0.8

53.9

0.5

54.0

0.7

51.9

1.1

0.14

Added sugar (% energy)

5.8

0.2

6.3

0.3

6.1

0.2

5.8

0.3

5.9

0.5

0.87

Alcohol (g/1000 kcal)

3.2

0.4

5.2

0.8

2.6

0.5

3.3

0.7

5.7

1.1

0.89

Total dietary fiber (g/1000 kcal)

6.9

0.1

6.5

0.2

6.5

0.1

6.3

0.2

6.0

0.3

<0.001

Vitamin A (µgRAE/1000 kcal)

400

11.3

414

19.2

377

12.3

378

18.6

334

29.0

0.021

Vitamin B1 (mg/1000 kcal)

0.44

0.01

0.43

0.01

0.43

0.01

0.42

0.01

0.42

0.01

0.002

Vitamin B2 (mg/1000 kcal)

0.75

0.01

0.74

0.02

0.73

0.01

0.72

0.02

0.69

0.03

0.026

Niacin (mgNE/1000 kcal)b

16.1

0.2

16.0

0.3

15.5

0.2

15.8

0.3

15.7

0.5

0.14

Vitamin B6 (mg/1000 kcal)

0.70

0.01

0.69

0.02

0.66

0.01

0.67

0.02

0.67

0.03

0.006

Vitamin B12 (µg/1000 kcal)

5.2

0.1

5.2

0.2

4.8

0.2

5.0

0.2

4.9

0.4

0.12

Folate (µg/1000 kcal)

189

3.6

180

6.1

176

3.9

170

5.9

164

9.3

<0.001

Vitamin C (mg/1000 kcal)

64.8

1.6

57.6

2.7

58.2

1.7

56.9

2.6

56.8

4.1

0.003

Vitamin D (mg/1000 kcal)

7.3

0.2

7.0

0.4

6.6

0.3

6.9

0.4

6.7

0.6

0.09

Sodium (mg/1000 kcal)

2344

27

2232

46

2288

29

2281

44

2244

69

0.15

Potassium (mg/1000 kcal)

1472

22

1407

37

1387

24

1356

36

1313

56

<0.001

Calcium (mg/1000 kcal)

318

5.8

307

10.0

302

6.4

293

9.6

273

15.1

0.001

Magnesium (mg/1000 kcal)

143

1.8

138

3.0

136

1.9

134

2.9

131

4.6

<0.001

Iron (mg/1000 kcal)

4.4

0.1

4.3

0.1

4.2

0.1

4.1

0.1

4.0

0.2

<0.001

Zinc (mg/1000 kcal)

4.5

0.03

4.4

0.1

4.4

0.04

4.4

0.1

4.3

0.1

0.11

Copper (mg/1000 kcal)

0.63

0.01

0.60

0.01

0.62

0.01

0.60

0.01

0.58

0.02

0.009

Diet quality score

NRF9.3

695

6.5

688

11.0

674

7.1

672

10.7

662

16.7

0.006

NRF9.3, Nutrient-Rich Food Index 9.3; SE, standard error; SFA, saturated fatty acid; RAE, retinol activity equivalent; NE, niacin equivalent.

All values were adjusted for age, education level, household income, living with children, residential area, and smoking status.

a A linear trend test was used with the median value in each category (0, 8, 25, 39, and 46 hours for 0 [housewives], 1–14, 15–34, 35–42, and ≥43 hours, respectively) as a continuous variable in the multivariable general linear regression.

b Niacin equivalents were calculated as niacin (mg) + protein (mg)/6000 according to the Dietary Reference Intakes for the Japanese, 2015 [25].

NRF9.3

695

6.5

688

11.0

674

7.1

672

10.7

662

16.7

0.006

NRF9.3, Nutrient-Rich Food Index 9.3; SE, standard error; SFA, saturated fatty acid; RAE, retinol activity equivalent; NE, niacin equivalent.

All values were adjusted for age, education level, household income, living with children, residential area, and smoking status.

a A linear trend test was used with the median value in each category (0, 8, 25, 39, and 46 hours for 0 [housewives], 1–14, 15–34, 35–42, and ≥43 hours, respectively) as a continuous variable in the multivariable general linear regression.

b Niacin equivalents were alculated as niacin (mg) + protein (mg)/6000 according to the Dietary Reference Intakes for the Japanese, 2015 [25].

Reviewer 2 Report

In the introduction, the research gap must be clearly defined.
The discussion should refer to the date of the research - 2013. What proves that the obtained research results are still valid? The timing of the research is one of the limitations.
You must specify the article contribution to the resources of science.
No recommendations for public policy entities, including public health.

Author Response

We appreciate all of your comments and suggestions. We found them quite useful during the revision.

Author's Reply to the Review Report (Reviewer 2)

Comment 1:

In the introduction, the research gap must be clearly defined.

Reply:

We have rewritten the Introduction to make the research gap clear according to your comment. In particular, we focused on two main points regarding the research gap: 1) diet is rarely comprehensively examined in this topic; and 2) no study has yet been conducted in the countries where women perform a large share of the housework (women spend over fivefold as much time as men). We hope this revision better explains the necessity of this study.

L43-84 (1. Introduction)

Married women have traditionally been responsible for much of the housework [1]. Throughout the last few decades, in addition to a role in housework, a growing number of married women have also engaged in paid work. For instance, in Japan, the employment rate of married women aged 25–64 years old has considerably increased (from 55% in 2000 to 71% in 2020) [2]. Still, the hours of unpaid work (e.g., housework and child care) taken up by women are so much longer than those by men globally [3,4].

Engagement in paid work results in less time for other activities. Previous studies on women and employment reported that their long hours of paid work or full-time work were associated with less time for meal preparation [5,6], which may lead to unhealthy dietary intake [5,7]. In contrast, engagement in paid work provides the benefit of income, which may improve diet quality [8]. Despite the increase in the employment rate of married women, few previous studies have examined the association of engagement in paid work with dietary habits among married women or women considered to have a role in unpaid work [5,9-13]. In particular, the overall dietary intake has not been well examined in relation to engagement in paid work [11,12]. Most previous studies have focused on limited dietary habits, such as breakfast habits [5,9], fast food consumption [5,10,13], and intake of select food groups or beverages [5,13]. Examining overall diet is beneficial to pro-vide useful indications for improving diet and health in married women who engage in paid work or who hold multiple roles.

The association of engagement of paid work with dietary habits can vary among the background of target populations. For women who have a large responsibility for meal preparation, for example, the impact of their hours of paid work on dietary behaviors can be substantial. According to the Global Gender Gap Report in 2018, women across the world spend twice as much time on average performing unpaid work (e.g., housework and child care) than men; however, women in Japan, Korea, and India spend over fivefold as much time [14]. In Japan, the average daily time spent on meal preparation and dishwashing is 142 minutes for wives and 6 minutes for husbands in households with housewives, whereas this time is 110 minutes for wives and 8 minutes for husbands in dual-earner households [15]. This finding suggests that women in dual-earner households have less time for meal preparation. In such countries where women have a large share of the housework, the impact of engagement in paid work on dietary behaviors can be substantial because married women in these countries are more likely to prepare meals. However, no study has yet examined the association between engagement in paid work and dietary intake among married women in the countries mentioned above except for a study on pregnant women with much lower employment rates [11]. The length of hours of paid work should also be considered, given that time scarcity has been reported to be negatively associated with some aspects of a healthy diet [16–20].

Two main points have been identified as a research gap: 1) diet is rarely comprehensively examined in this subject; and 2) no study has been conducted to date in the countries where women perform a large share of the housework. This study aimed to investigate the association of hours of paid work with dietary intake and diet quality among married women in Japan, where women assume a large share of the housework. We hypothesized that the dietary intake of married women with longer hours of paid work is less healthy than that of those with shorter hours of paid work (including housewives).

[2] Statistics Bureau, Ministry of Internal Affairs and Communications. Labor Force Survey Basic Tabulation Whole Japan Yearly, Table 1-4-1, Population of 15 years old and over by labor force status, marital status and age groups. Available online: https://www.e-stat.go.jp/en/dbview?sid=0003008335 (accessed on Aug 15, 2021).

[8] Darmon, N.; Drewnowski, A. Does social class predict diet quality? Am. J. Clin. Nutr. 2008, 87, 1107-1117.

[15] Statistics Bureau, Ministry of Internal Affairs and Communications. Survey on Time Use and Leisure Activities 2016 Survey on Time Use and Leisure Activities Questionnaire B Results on Time Use by Detailed Activity Coding Time Use, Table 16-1, Average time spent in activities for all persons (Main Activities) by Kind of activities, Day of the week, Sex, Family type of household, Usual economic activities of a married couple (Husbands and Wives)-Japan. Available online: https://www.e-stat.go.jp/en/dbview?sid=0003213070 (accessed on Aug 15, 2021).

Comment 2:

The discussion should refer to the date of the research - 2013. What proves that the obtained research results are still valid? The timing of the research is one of the limitations.

Reply:

Thank you for making this important point. Because it is difficult to concluded what changes may have occurred solely based on this study, we have added this point as one of the limitations.

According to the National Labor Force Survey, the employment rate of married women aged 25−54 years has increased from 65% in 2013 to 74% in 2020 [2] in Japan. Little has changed in the amount of time that husbands and wives spent on meal preparation and dishwashing between 2011 and 2016 according to the results of National Time Use surveys (men, 7 minutes in 2011 and 8 minutes in 2016; women, 112 minutes in 2011 and 110 minutes in 2016) [15, 46]. The latest data available were for 2016 because this survey is conducted every 5 years, but we assumed that there has been only a slight change in these indicators. We have included this information in the Limitations section.

L386-394 (4. Discussion)

Second, this study was conducted in 2013, and since then the association between hours of paid work and dietary intake for married women may have been changed depending on their social circumstances. Little has changed in the amount of time that married men and women spend on meal preparation and dishwashing between the 2011 [46] and 2016 surveys [15], whereas the employment rate of married women age 15-64 years has increased from 64% in 2013 to 74% in 2020 [2]. It is difficult to conclude solely based on the findings of this study what impact these changes may have had on the association of hours of paid work with dietary intake in married women. Future studies are needed to examine this association at periodic intervals.

[2] Statistics Bureau, Ministry of Internal Affairs and Communications. Labor Force Survey Basic Tabulation Whole Japan Yearly, Table 1-4-1, Population of 15 years old and over by labor force status, marital status and age groups. Available online: https://www.e-stat.go.jp/en/dbview?sid=0003008335 (accessed on Aug 15, 2021).

[15] Statistics Bureau, Ministry of Internal Affairs and Communications. Survey on Time Use and Leisure Activities 2016 Survey on Time Use and Leisure Activities Questionnaire B Results on Time Use by Detailed Activity Coding Time Use, Table 16-1, Average time spent in activities for all persons (Main Activities) by Kind of activities, Day of the week, Sex, Family type of household, Usual economic activities of a married couple (Husbands and Wives)-Japan. Available online: https://www.e-stat.go.jp/en/dbview?sid=0003213070 (accessed on Aug 15, 2021).

[46] Statistics Bureau, Ministry of Internal Affairs and Communications. Survey on Time Use and Leisure Activities 2011 Survey on Time Use and Leisure Activities Questionnaire B Results on Time Use Time Use, 0180202 Average Time Spent for All Persons, for Participants and Participation Rate in Main Activities and Simultaneous Activities (Minor Groups) by Day of the Week, Family Type of Household and Usual Economic Activities of a Married Couple (Husbands and Wives). Available online: https://www.e-stat.go.jp/en/dbview?sid=0003070457 (accessed on Aug 15, 2021).

Comment 3:

You must specify the article contribution to the resources of science.

Reply:

According to your comment, we have rewritten the following sentence to specify that this article examined the overall diet and identified challenges that married women may encounter when they engage in paid work. We have also added a paragraph in the Discussion section to specify the contribution to the literature.

L317-332 (4. Discussion)

Monetary affluence is likely one of the factors related to a healthy diet [8], which may explain some of the association between hours of paid work and dietary behaviors. In a study in Pakistan of women with children, working women had a higher intake of fruits and vegetables compared with housewives [13]. This finding is perhaps because working women had a much higher household income than housewives in that population. In contrast, in a US study, women with full-time work had less fruit and vegetable intake than housewives, despite higher household income among women with full-time work than housewives [5]. That study also showed less food preparation time among women with full-time work compared with housewives, which indicates that food preparation time may have had a large impact on their dietary intake rather than income in that population. In the current study, hours of paid work were associated with unhealthy dietary intake, independent of household income. It is possible that time constraints due to paid work may affect dietary intake in Japanese married women. Nevertheless, the association between hours of paid work and dietary habits may vary based on the sociodemographic characteristics of the target population, and considering these factors is needed for thorough understanding of the association of hours of paid work with dietary intake.

L341-351 (4. Discussion)

We observed negative associations with hours of paid work of intake of dietary fiber and many vitamins and minerals. Lower intakes of these nutrients can be attributed to mainly lower intake of vegetables and, in part, lower intake of potatoes and pulses. In contrast, rice was positively associated with hours of paid work in this study. In previous Japanese studies, a higher rice intake was associated with lower intakes of other food groups (e.g., vegetables), dietary fiber, and micronutrients [43,44]. Considering that rice is a major staple food in Japan and relatively energy-dense, a higher rice intake co-occurring with a lower micronutrient intake is unsurprising. By examining overall dietary intake, the current study identified foods and nutrients that were more versus less affected by engagement in paid work. Married women’s engagement in paid work may affect their intake of healthy foods and nutrients, not unhealthy ones, in Japan.

L418-425 (5. Conclusion)

This study showed that married women engaging in paid work, especially those with long hours of paid work, have lower dietary quality, lower intake of vegetables, soy products, seaweeds, and potatoes, as well as lower intakes of dietary fiber, and many vitamins and minerals in Japan. These findings suggest that hours of paid work are one of the factors that led to unhealthy dietary intake in married women in context of these women assuming a large share of the housework. Although housewives also have room for improvement in their dietary intake, further research is needed to determine what factors are necessary for married women who engage in paid work to facilitate healthy dietary intake.

Comment 4:

No recommendations for public policy entities, including public health.

Reply:

We have added the following sentences as recommendations for public policy entities, in accordance with your comment.

L366-376 (4. Discussion)

The current findings suggest the necessity of establishing an environment where married women who engage in paid work can improve their dietary intake in Japan. Although further research is needed, public policy efforts are also required to establish such an environment. Increasing the availability of healthy and less time-consuming foods may be one of the useful strategies for coping with a lack of time for food preparation [17,45]. Additionally, regardless of sex, it may be essential to control the appropriate hours of paid work. Japanese men spend longer hours in paid work than men in any other OECD country [4], which may make it difficult for men to share the housework with their partners. Although it is not possible to provide effective interventions only based on the current findings, public health strategies may be required to not only encouraging individual efforts for improving their dietary intake but also focus on decreasing time burdens among workers.

[17] Horning, M.L.; Fulkerson, J.A.; Friend, S.E.; Story, M. Reasons Parents Buy Prepackaged, Processed Meals: It Is More Complicated Than “I Don't Have Time”. J. Nutr. Educ. Behav. 2017, 49, 60–66.

[45] Devine, C.M.; Farrell, T.J.; Blake, C.E.; Jastran, M.; Wethington, E.; Bisogni, C.A. Work conditions and the food choice coping strategies of employed parents. J. Nutr. Educ. Behav. 2009, 41, 365–370.

L418-425 (5. Conclusion)

This study showed that married women engaging in paid work, especially those with long hours of paid work, have lower dietary quality, lower intake of vegetables, soy products, seaweeds, and potatoes, as well as lower intakes of dietary fiber, and many vitamins and minerals in Japan. These findings suggest that hours of paid work are one of the factors that led to unhealthy dietary intake in married women in context of these women assuming a large share of the housework. Although housewives also have room for improvement in their dietary intake, further research is needed to determine what factors are necessary for married women who engage in paid work to facilitate healthy dietary intake.

While revising, we noticed that some values in Table 3 were incorrect. We have corrected this error (red font).

Table 3. Nutrient intake and NRF9.3 score according to the categories of hours of paid work in 644 Japanese married women.

Hours of paid work per week

0
(Housewive)s

1–14

15–34

35–42

≥43

pa

(n = 240)

(n = 83)

(n = 198)

(n = 88)

(n = 36)

Mean

SE

Mean

SE

Mean

SE

Mean

SE

Mean

SE

Nutrients

Protein (% energy)

15.5

0.2

15.2

0.3

15.0

0.2

15.1

0.3

14.7

0.4

0.027

Total fat (% energy)

28.5

0.3

28.6

0.5

28.1

0.3

27.3

0.5

28.2

0.8

0.10

SFA (% energy)

7.9

0.1

8.1

0.2

7.9

0.1

7.5

0.2

7.8

0.3

0.27

Carbohydrate (% energy)

52.8

0.4

51.5

0.8

53.9

0.5

54.0

0.7

51.9

1.1

0.14

Added sugar (% energy)

5.8

0.2

6.3

0.3

6.1

0.2

5.8

0.3

5.9

0.5

0.87

Alcohol (g/1000 kcal)

3.2

0.4

5.2

0.8

2.6

0.5

3.3

0.7

5.7

1.1

0.89

Total dietary fiber (g/1000 kcal)

6.9

0.1

6.5

0.2

6.5

0.1

6.3

0.2

6.0

0.3

<0.001

Vitamin A (µgRAE/1000 kcal)

400

11.3

414

19.2

377

12.3

378

18.6

334

29.0

0.021

Vitamin B1 (mg/1000 kcal)

0.44

0.01

0.43

0.01

0.43

0.01

0.42

0.01

0.42

0.01

0.002

Vitamin B2 (mg/1000 kcal)

0.75

0.01

0.74

0.02

0.73

0.01

0.72

0.02

0.69

0.03

0.026

Niacin (mgNE/1000 kcal)b

16.1

0.2

16.0

0.3

15.5

0.2

15.8

0.3

15.7

0.5

0.14

Vitamin B6 (mg/1000 kcal)

0.70

0.01

0.69

0.02

0.66

0.01

0.67

0.02

0.67

0.03

0.006

Vitamin B12 (µg/1000 kcal)

5.2

0.1

5.2

0.2

4.8

0.2

5.0

0.2

4.9

0.4

0.12

Folate (µg/1000 kcal)

189

3.6

180

6.1

176

3.9

170

5.9

164

9.3

<0.001

Vitamin C (mg/1000 kcal)

64.8

1.6

57.6

2.7

58.2

1.7

56.9

2.6

56.8

4.1

0.003

Vitamin D (mg/1000 kcal)

7.3

0.2

7.0

0.4

6.6

0.3

6.9

0.4

6.7

0.6

0.09

Sodium (mg/1000 kcal)

2344

27

2232

46

2288

29

2281

44

2244

69

0.15

Potassium (mg/1000 kcal)

1472

22

1407

37

1387

24

1356

36

1313

56

<0.001

Calcium (mg/1000 kcal)

318

5.8

307

10.0

302

6.4

293

9.6

273

15.1

0.001

Magnesium (mg/1000 kcal)

143

1.8

138

3.0

136

1.9

134

2.9

131

4.6

<0.001

Iron (mg/1000 kcal)

4.4

0.1

4.3

0.1

4.2

0.1

4.1

0.1

4.0

0.2

<0.001

Zinc (mg/1000 kcal)

4.5

0.03

4.4

0.1

4.4

0.04

4.4

0.1

4.3

0.1

0.11

Copper (mg/1000 kcal)

0.63

0.01

0.60

0.01

0.62

0.01

0.60

0.01

0.58

0.02

0.009

Diet quality score

NRF9.3

695

6.5

688

11.0

674

7.1

672

10.7

662

16.7

0.006

NRF9.3, Nutrient-Rich Food Index 9.3; SE, standard error; SFA, saturated fatty acid; RAE, retinol activity equivalent; NE, niacin equivalent.

All values were adjusted for age, education level, household income, living with children, residential area, and smoking status.

a A linear trend test was used with the median value in each category (0, 8, 25, 39, and 46 hours for 0 [housewives], 1–14, 15–34, 35–42, and ≥43 hours, respectively) as a continuous variable in the multivariable general linear regression.

b Niacin equivalents were calculated as niacin (mg) + protein (mg)/6000 according to the Dietary Reference Intakes for the Japanese, 2015 [25].

Round 2

Reviewer 1 Report

Thank you for the thorough and detailed revision of the article. In the presented form, the work is comprehensive, has more significant scientific impact.